



# Two-dimensional impurity imaging in deep Antarctic ice cores: Snapshots of three climatic periods and implications for high-resolution signal interpretation

Pascal Bohleber[1], Marco Roman[1], Martin Šala[2], Barbara Delmonte[3], Barbara Stenni[1], and Carlo Barbante[1,4]

[1]Department of Environmental Sciences, Informatics and Statistics, Ca'Foscari University of Venice, Italy
[2]Department of Analytical Chemistry, National Institute of Chemistry, Ljubljana, Slovenia
[3]Università degli Studi di Milano-Bicocca, Dept. of Earth and Environmental Sciences, Milano, Italy
[4]Institute of Polar Sciences, CNR, Venice, Italy

**Correspondence:** Pascal Bohleber (pascal.bohleber@unive.it)

**Abstract.** Due to its micron-scale resolution and micro-destructiveness, laser ablation inductively coupled plasma mass spectrometry (LA-ICP-MS) is especially suited for exploring closely spaced layers in the oldest and highly thinned sections of polar ice cores. Recent adaptions of the LA-ICP-MS technique have achieved fast washout times as the basis for introducing state-of-the-art 2D imaging to ice core analysis. This new method has great potential in its application for investigating the

localization of impurities on the ice sample, crucial to avoid misinterpretation of ultra-fine resolution signals. Here first results are presented from applying LA-ICP-MS elemental imaging to selected glacial and interglacial samples of the Talos Dome and EPICA Dome C ice cores from central Antarctica. The localization of impurities with both marine and terrestrial sources is discussed, revealing generally a strong connection with the network of grain boundaries but also distinct differences among climatic periods. Scale-dependent image analysis shows that the spatial significance of a single line profile along the main

core axis increases systematically as the imprint of grain boundaries weakens. With this, it is demonstrated how instrumental settings can be adapted specifically fit-for-purpose, i.e. either to employ LA-ICP-MS to study the impurity-microstructure interplay or to investigate highly thinned climate proxy signals in deep polar ice.

## 1 Introduction

Antarctic ice cores are one of the cornerstones in today's paleoclimate research, archiving a unique variety of proxies such

as greenhouse gas and aerosol-related atmospheric impurity records over time-scales from decades to hundreds of millennia (e.g. Petit et al., 1999; Kawamura et al., 2003; Ahn et al., 2004; EPICA Community Members, 2004). The investigation of the oldest, deepest and highly thinned ice core layers has become of special interest in state-of-the-art polar ice core research, in particular regarding a 1.5 million year-old record to be recovered from Antarctica (Brook et al., 2006; Fischer et al., 2013). With each meter of the deep ice core section expected to comprise more than 10,000 years (Lilien et al., 2020), there is a demand

for new analytical methods that surpass the resolution capabilities of established methods based on meltwater analysis, such as continuous flow analysis (CFA) (e.g. Röthlisberger et al., 2000; McConnell et al., 2002; Osterberg et al., 2006; Kaufmann



et al., 2008). In this framework, laser ablation inductively coupled plasma mass spectrometry (LA-ICP-MS) has recently been re-established for high-resolution trace element characterization of ice cores (Müller et al., 2011; Sneed et al., 2015). The comparison of the novel LA-ICP-MS signals with CFA revealed that the low frequency variability seen in LA-ICP-MS signals

is consistent with the full resolution impurity records obtained by CFA (Della Lunga et al., 2017; Spaulding et al., 2017). Initial investigations were also performed regarding the relationship between LA-ICP-MS signals and abundantly observed ice crystal features such as grain boundaries and triple junctions (Della Lunga et al., 2014, 2017; Kerch, 2016; Beers et al., 2020). Advances through employing ablation cells dedicated to fast washout as well as optimized lasing and ICP-MS settings have introduced a new state-of-the-art in imaging techniques with LA-ICP-MS (Wang et al., 2013; van Elteren et al., 2019). Recently,

this new approach was transferred to ice core analysis with LA-ICP-MS, now offering to study the impurity distribution in ice cores in 2D. First results clearly demonstrated a close connection between grain boundaries and high-resolution LA-ICP-MS signals, which became particularly evident for the example of sodium (Bohleber et al., 2020).

As a result of these developments, LA-ICP-MS promises a tool for investigating the localization of impurities in the ice matrix, supplementing methods such as scanning electron microscopy equipped with energy-dispersive X-ray spectroscopy (SEM-

EDS) (Barnes et al., 2003; Barnes and Wolff, 2004; Iliescu and Baker, 2008) and micro-Raman spectroscopy (Sakurai et al., 2011; Eichler et al., 2017). Understanding micro-scale impurity localization is not only significant regarding the deformational (Dahl-Jensen et al., 1997) and dielectric properties (Stillman et al., 2013) of glacier ice on a macroscopic scale. It also becomes of special importance to assess the stratigraphic integrity of high-resolution impurity records in deep and comparatively warm ice, typically characterized by large grains (Faria et al., 2014a). Especially for soluble species, the advection of impurity

anomalies through diffusion along the grain boundary network has been discussed as a possible cause of post-depositional alteration of ice core paleoclimate records (Rempel et al., 2001).

The impurity localization at grain boundaries also has direct consequences with respect to the interpretation of 1D LA-ICP-MS profiles, i.e. single lines measured along the main core axis. Such line profiles have been used to obtain a high-resolution time series of paleoclimate signals, after further smoothing (e.g. Mayewski et al., 2014; Haines et al., 2016; Della Lunga et al.,

2017). It has been suggested that, in presence of signal imprints related to the ice crystal matrix, it is beneficial to average the signal between two or more parallel tracks (Della Lunga et al., 2017). An according standard technique has not yet been developed, however. Evidently, the need to evaluate signal reproducibility in parallel lines will depend on local variations in grain sizes and in the imprint of grain boundaries. In this context, 2D imaging can provide a more refined approach to assess the origin of LA-ICP-MS signal and the spatial significance of single line profiles.

Employing the recently developed LA-ICP-MS imaging technique (Bohleber et al., 2020) selected ice samples of various depth sections representative of distinct climatic periods in Antarctic ice cores were analyzed, aiming to include a broad spectrum of ice properties, such as age and mean grain size. This captured snapshots of the 2D impurity distribution as seen by LA-ICP-MS elemental imaging, which provides additional important detail regarding the localization of impurities in relation to the grain boundary network. In presence of a variable signal imprint from the grain boundaries, the spatial significance of a single line

profile along the main core axis is carefully assessed based on 2D images. It is shown how measurement settings can be adapted





on this ground in order to specifically employ LA-ICP-MS line profiles to investigate highly thinned climate proxy signals in deep polar ice.

## 2 Materials and Methods

The LA-ICP-MS setup employed at the University of Venice comprises an Analyte Excite ArF* excimer 193 nm laser (Teledyne CETAC Photon Machines) and an iCAP-RQ quadrupole ICP-MS (Thermo Scientific). The laser ablation system includes a HelEx II 2-volume ablation cell mounted on a high-precision xy-stage and equipped with a custom-build cryogenic sample holder in order to permit the analysis of ice core samples below freezing temperatures. The circulation of a glycol–water mixture cooled to $-30\,°C$ keeps the ice samples surface temperature durably at $-18\pm2\,°C$. Established procedures for ice sample preparation and surface decontamination in LA-ICP-MS ice core analysis were followed (Della Lunga et al., 2014, 2017). A band saw was used in a cold-room ($-20\,°C$) to cut ice samples to a strip geometry (9 x 2 cm). Ice thickness was then reduced to 1.5 cm by manual scraping using a custom-built PTFE vice with a ceramic $ZrO_2$ blade (American Cutting Edge, USA).

Helium was used as a carrier gas for aerosol transport from the sample surface to the ICP-MS. As an important addition, ARIS, a rapid aerosol transfer line, was integrated, resulting in a washout time of ~34 ms. This refers to the time needed for transferring the ablated sample aerosol plume to the ICP-MS, which is principally determined by the extraction efficiency from the ablation cell and subsequent dispersion into the transfer line. With washout times in the tens of ms range, the recording of baseline-separated single pulses at high repetition rates becomes possible; 294 Hz and a dosage of 10 were used here. This allows scanning of the surface at around one millimeter per second, which is roughly 10 times faster than previous studies on ice cores (Della Lunga et al., 2017; Spaulding et al., 2017). As a result, artifact-free elemental maps can be generated at high spatial resolution in a limited run time and/or over comparatively large areas.

First, at least one full pre-ablation run was conducted for further decontamination, using a spot size of 150 mm square as well as bidirectional scanning. Next, the maps are obtained as a pattern of lines with a 35 $\mu$m square spot size, fluence 3.5 J/cm$^2$, without overlap perpendicular to the scan direction, and without any post-acquisition spatial interpolation. Lateral resolution is 35 $\mu$m both along and perpendicular to the scan direction. Due to the precise synchronization of data acquisition required to avoid image artifacts, the number of analytes/isotopes is restricted. Considered in the following are $^{23}$Na, $^{25}$Mg and $^{88}$Sr. ICP-MS dwell times were 4, 4.6 and 10 ms, respectively. The choice of these analytes was primarily due to their significance as a paleoclimate proxy in polar ice cores (Legrand and Mayewski, 1997): Na being related mostly to sea-salt, Mg with both marine and terrestrial sources and Sr as a chemically similar substitute for Ca, mostly related to terrestrial sources. Ca is analytically challenging in ICP-MS due to significant spectral interferences. Scan lines on a NIST glasses SRM 612 and 614 were measured before and after the acquisition of each map. Background and drift correction as well as image construction were performed using the software HDIP (Teledyne Photon Machines, Bozeman, MT, USA).

Using the integrated camera co-aligned with the laser, it is possible to obtain a mosaic of optical images of the ice sample surface. Here, it becomes possible to see entrapped air bubbles (dark circles) and to distinguish individual ice crystals and their boundaries (dark lines). Comparing such optical images with the LA-ICP-MS elemental maps allows a clear assessment of the





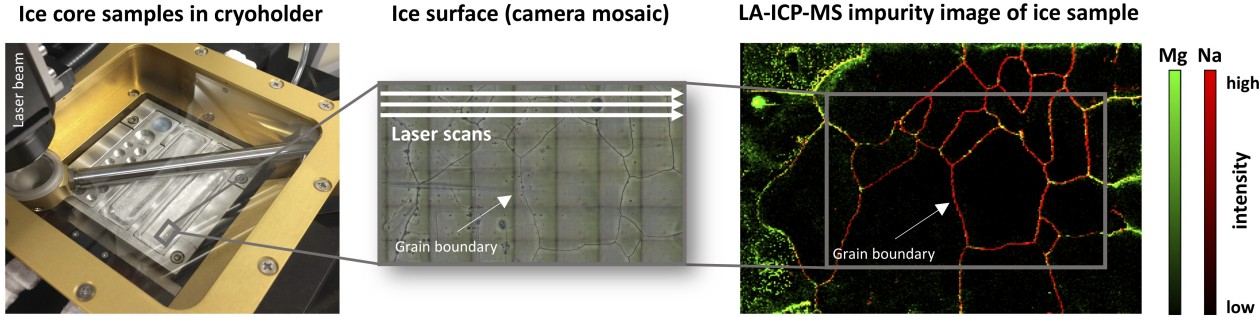

**Figure 1.** Generating 2D impurity images of ice cores by LA-ICP-MS. Strips of ice (approx. 9 x 2 cm, 1.5 cm thickness) are kept frozen in a custom-designed cryogenic holder (left panel). A mosaic of optical images allows localizing ice surface features such as grain boundaries (middle panel). Data from an array of non-overlapping laser scan lines are stacked without interpolation to generate an elemental map of impurities distribution (right panel).

**Table 1.** Samples from Antarctic ice cores analyzed by LA-ICP-MS impurity imaging.

| Ice core | Bag nr. | Age (approx. ka BP) | Climatic period | Image size (mm) |
|---|---|---|---|---|
| Talos Dome | 375-B1 | 5 | Holocene | 7 x 31.5 |
| EPICA Dome C | 1065 | 27.8 | MIS 2 | 7 x 35.0 |
| EPICA Dome C | 3092 | 129 | MIS 5.5 | 10.5 x 18.5 |

localization of impurities within the ice crystal matrix (Figure 1). Further details of the ice core impurity imaging method have
90    already been described elsewhere (Bohleber et al., 2020).

Sample selection was guided to consider ice of various depth sections and climatic periods in Antarctic ice cores, including glacial as well as interglacial periods. At the same time, the deepest sections were still avoided since featuring very large grain size calling for mapping large areas (see Discussion). For the purpose of this study, the analysis focused on three exemplary datasets with the highest image quality (Table 1). The Talos Dome Holocene ice sample (referred to in the following as TD

95    Holocene) is from a depth section featuring an average grain size of 1 – 2 mm (Montagnat et al., 2012). Likewise, the EPICA Dome C core (EDC) at a depth of 585.2 m corresponds to Marine Isotope Stage 2 (EDC MIS 2) and has an average grain radius around 1.5 mm. Notably, this sample is also from a section characterized by a very high dust content (average around 500 $\mu$g kg$^{-1}$). In contrast, the sample from MIS 5.5 (EDC MIS 5.5) is from 1700.5 m depth, characterized by low dust levels but a local maximum in grain radius, around 3.5 mm (EPICA Community Members, 2004).



## 3 Results

### 3.1 Basic elemental maps in comparison with optical images

The obtained maps of elemental intensity distribution are shown in Figures 2, 3 and 4, together with the optical images of the corresponding sample surface. All three analytes typically show sufficiently high signal/noise ratios. The three sets of maps show clear differences but are, at first order, composed of similar basic features. Sorted by increasing spatial extent, the basic features are: i) individual bright spots, typically comprising just a few clustered bright pixels, ii) a network of lines, dominant for the Na maps, iii) mm-scale differences in the intensity, some sections being distinctly lower in intensity with respect to the others. The comparison with the optical images clearly shows that the network of high-intensity lines can be associated with the location of grain boundaries. For the individual bright spots, candidates are dust particles and micro-inclusions, however finding a clear association with optical features is generally difficult. In all their maps, Mg and Sr show a certain degree of similarity in spatial distribution, but clear differences with Na. In more detail, among the maps/images the following is observed:

- **Talos Dome, Holocene** (Figure 2): high Na intensities track the grain boundaries, with only a minor association with grain boundaries for Mg and Sr on the left map side. Mg and Sr show generally more distributed intensities with occasional bright spots and a bright stripe within the left center half of the map showing in all elements. Notably the latter has no clear counterpart in the optical image.

- **EPICA Dome C, MIS 2** (Figure 3): the sample is characterized by comparatively smaller grains, as expected in a glacial period (Gow et al., 1997; Thorsteinsson et al., 1997). Bright spots are abundant for all elements. The association of network lines with grain boundaries is again most clear for Na but also found in some regions for Mg and Sr. The latter also show high intensities in some grain interiors. Intensities are generally lower in the center and towards the left-hand edge of the map.

- **EPICA Dome C, MIS 5.5** (Figure 4): The sample stands out by showing a high degree of localization at grain boundaries for all elements. In-grain intensities of Mg and Sr are occasionally visible in vicinity of the boundaries. Bright spots are almost completely absent.

### 3.2 Impurity localization and co-localization analysis

It is important to note that the focus here lies on a relative comparison of the degree of co-localization among elements and between optical features, instead of an accurate quantification of co-localization. To this end, imaging the localization of impurities does not require a fully quantitative method, although recent advances have been made in the challenge of a matrix-matched calibration for LA-ICP-MS with artificial ice standards (Della Lunga et al., 2017). Basic co-localization analysis was performed in order to further investigate differences in Na maps with respect to Mg and Sr, and to detect potential minor differences in Mg and Sr signals distribution. As a first step to visual co-localization, a simple overlay composite image of the different elemental maps is included in Figures 2, 3 and 4. In the composite, each individual map is represented by a separate



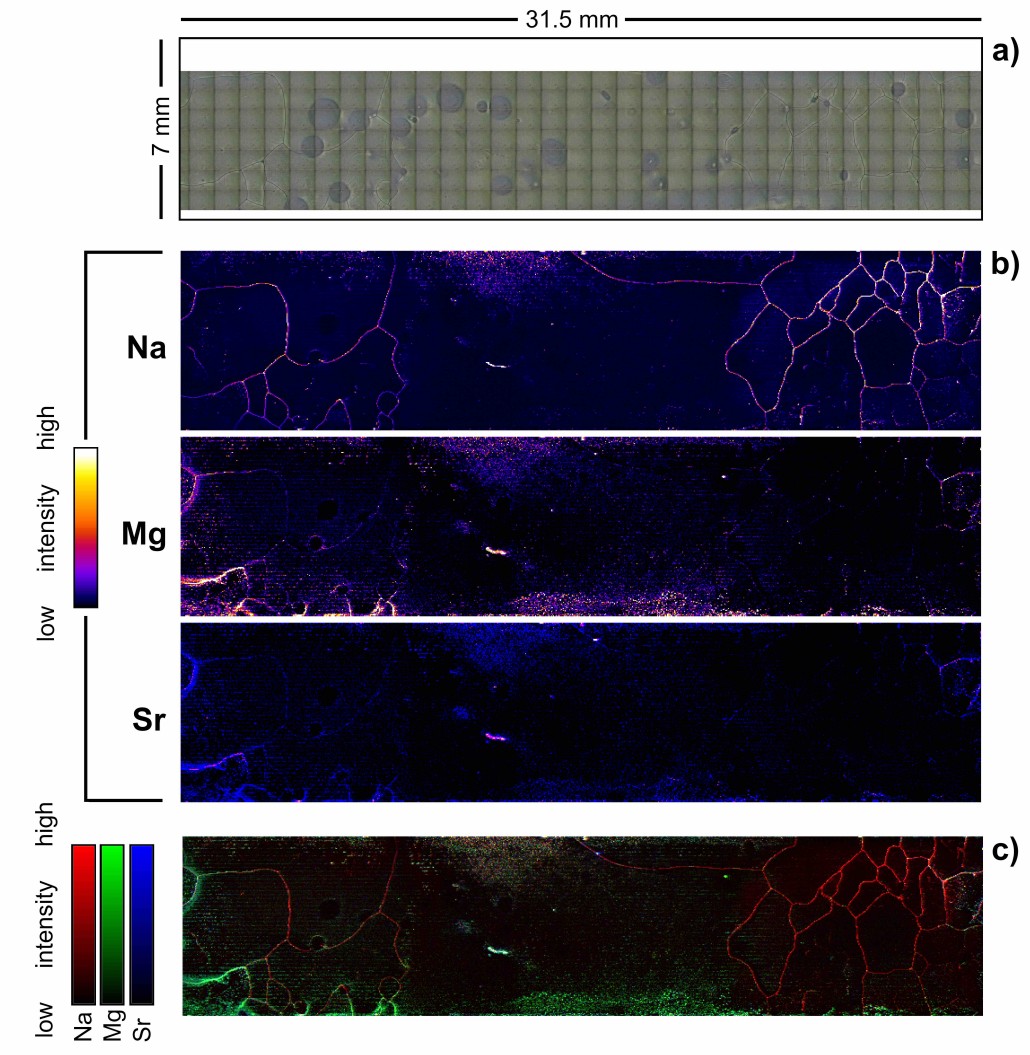

**Figure 2.** Talos Dome 375-B1 Holocene sample. Panel a) shows an optical image; Panel b) elemental maps of Na, Mg and Sr, respectively, in the same area; c) composite RGB map of the three elements. The main core axis runs from left (top) to right.

color channel, using red, green and blue for Na, Mg and Sr, respectively. This means that co-localized Na and Mg (green and red) will result in yellow hotspots whereas white areas represent all elements co-localized together. A preliminary visual analysis of the composite map already suggests that not all bright spots are co-localized, especially for the image of Figure 2 (MIS 2) where single channel bright spots stand out abundantly. However, the visual overlay has its limits regarding analyzing

co-localizations. Differences in absolute signal intensity and signal/noise ratio can subdue or mask co-localization.

For further comparison of the degree of co-localization, the matrices of intensity values that underlie the images shown in Figure 2, 3 and 4 were used to make scatterplots for each pair of elements. As becomes evident from Figure 5, the range in



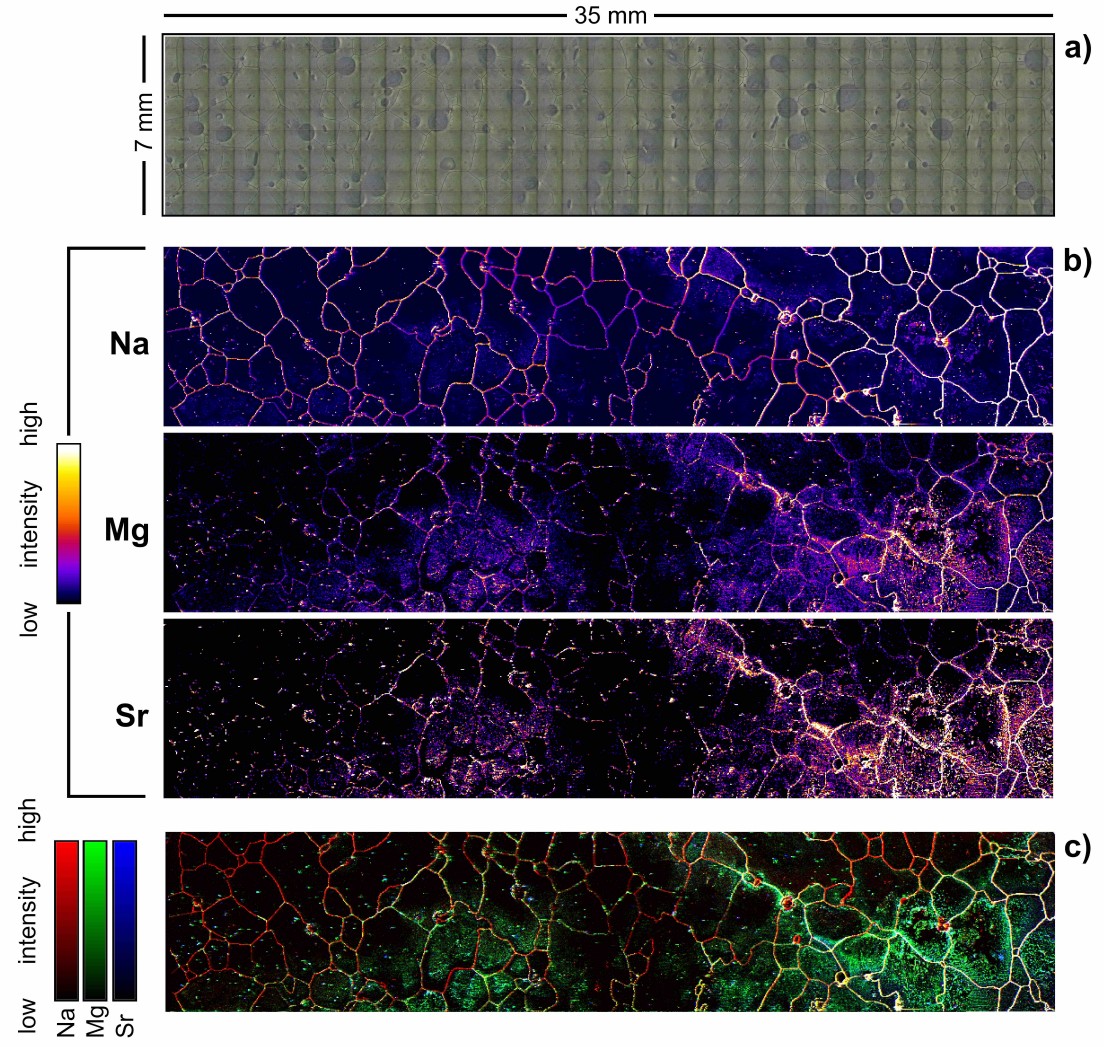

**Figure 3.** EPICA Dome C 1065 sample from MIS2. Panel a) shows an optical image; Panel b) elemental maps of Na, Mg and Sr, respectively, in the same area; c) composite RGB map of the three elements. The main core axis runs from left (top) to right.

intensities for Mg and Sr is generally similar, while Na intensities can be higher by several orders of magnitude. The relative higher background level seen in Na has been observed before in LA-ICP-MS ice core analysis and was suggested to be related to the use of NIST glasses as reference materials (Della Lunga et al., 2017). The scatterplots also indicate the almost absent co-localization in the TD Holocene image, showing signs of mutual exclusions (values extremely close to one axis). For EDC MIS2 some correlation emerges but becomes most evident for EDC MIS 5.5.

Several metrics to quantify co-localization in images, e.g. for tissue analysis with fluorescence microscopy, have been established in the literature (Bolte and Cordelières, 2006). For intensity correlation coefficient-based analyses, the Pearson's



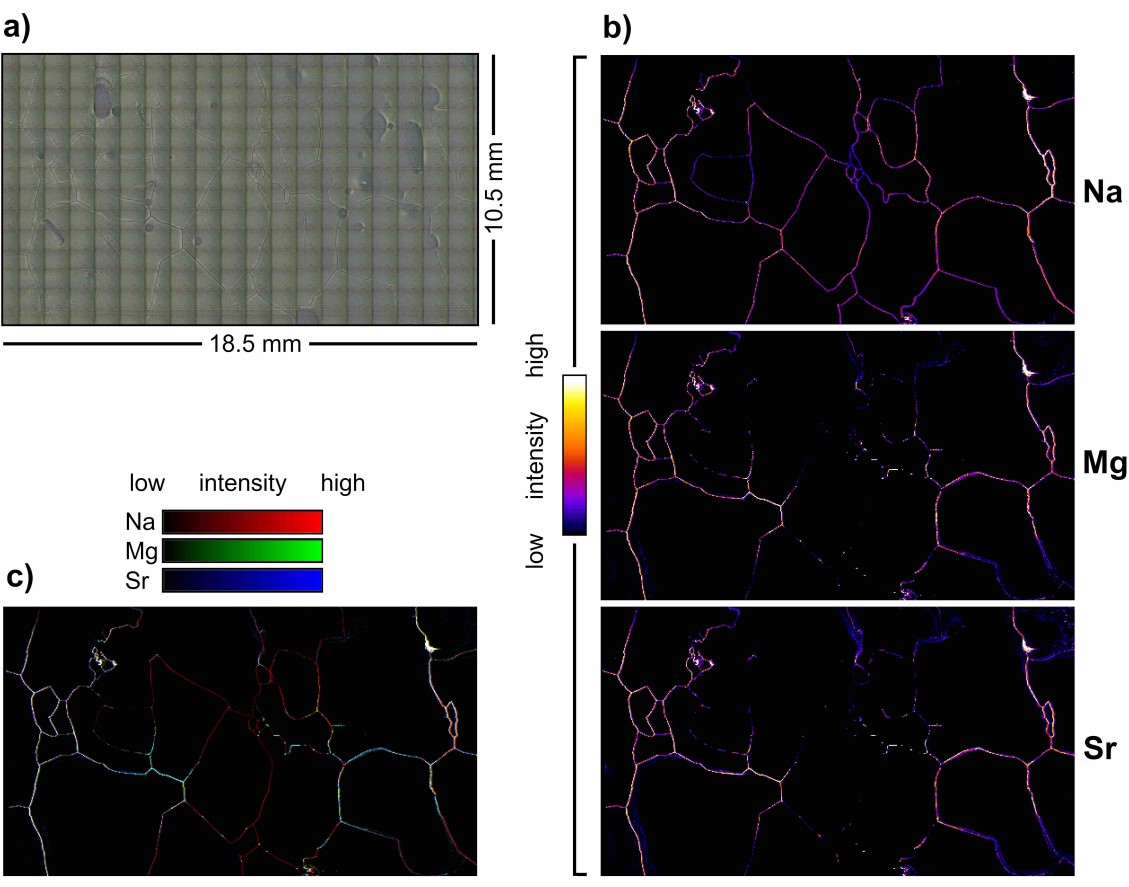

**Figure 4.** EPICA Dome C 3092 sample from MIS 5.5. Panel a) shows an optical image; Panel b) elemental maps of Na, Mg and Sr, respectively, in the same area; c) composite RGB map of the three elements. The main core axis runs from left (top) to right.

Correlation Coefficient (PCC), the Spearman's Rank Coefficient (SRC) and the Intensity Correlation Quotient (ICQ) were considered in this work, since superior in such cases over the Mander's Overlap Coefficient (Adler and Parmryd, 2010). The SRC equals the PCC applied to ranked data, in which case the relative amplification in Na intensities should not play a role. Since PCC and SRC can be sensitive to thresholding, the robustness of the values was checked against results obtained after discarding the 0.5 and 99.5 percentiles of the data. Compared to PCC and SRC, the ICQ has a higher sensitivity to midrange

pixels (Adler and Parmryd, 2010). It only considers the sign, or respectively whether each of the two intensities are above or below their respective mean intensity. By construction it ranges from -0.5 to 0.5, denoting negative and positive correlation, respectively (Li et al., 2004). The ICQ was calculated with the public domain tool named JACoP (Bolte and Cordelières, 2006) available for the software imageJ after converting the maps to grayscale.

  The results of co-localization analysis are summarized in Table 2. Regarding the comparison among the three samples, the

PCC, SRC and ICQ substantiate the previously noted findings from visual analysis. No clear evidence for inter-element corre-





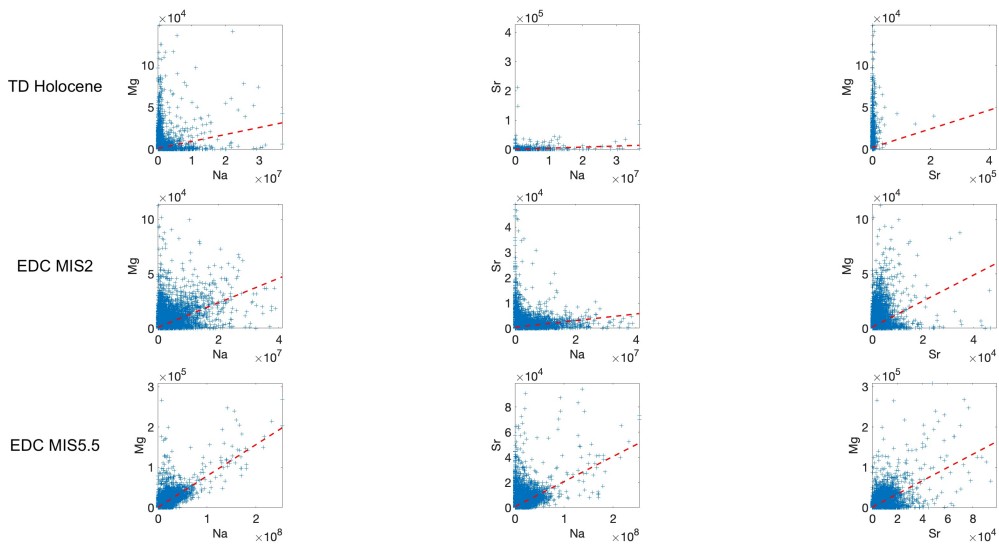

**Figure 5.** Scatterplots compiled from the matrices of intensity values (counts per second) underlying the images shown in Figure 2, 3 and 4. The result of applying linear regression (red dashed line) is shown purely for visual aid.

**Table 2.** Overview on results from elemental co-localization analysis using Pearson's correlation coefficient (PC), Spearman's Rank Coefficient (SRC) and the Intensity Correlation Quotient (ICQ). Values in parenthesis refer to results after discarding the 0.5 and 99.5 percentiles (see text).

| | Mg vs Na | | | Sr vs Na | | | Mg vs Sr | | |
|---|---|---|---|---|---|---|---|---|---|
| | PCC | SRC | ICQ | PCC | SRC | ICQ | PCC | SRC | ICQ |
| TD Holocene | 0.24 (0.18) | 0.18 (0.17) | 0.19 | 0.09 (0.15) | 0.32 (0.31) | 0.22 | 0.13 (0.34) | 0.29 (0.28) | 0.24 |
| EDC MIS 2 | 0.46 (0.44) | 0.41 (0.40) | 0.24 | 0.16 (0.23) | 0.25 (0.24) | 0.27 | 0.36 (0.46) | 0.49 (0.48) | 0.28 |
| EDC MIS 5.5 | 0.79 (0.67) | 0.62 (0.60) | 0.46 | 0.50 (0.34) | 0.55 (0.52) | 0.44 | 0.64 (0.62) | 0.60 (0.60) | 0.46 |

lation is found for the TD Holocene sample. In contrast, correlation is generally strongest for the EDC MIS5.5 sample, with the highest PCC of 0.79 for Na and Mg. The EDC MIS2 sample presents an intermediate case.

## 3.3 An exploration of analysis by image segmentation

The fact that Na shows a clear signal at all grain boundaries allows to perform image segmentation based on the LA-ICP-MS
alone, without using the optical mosaics which lack sufficient image quality. Segmentation means here to extract the position of all pixels in the image that are part of the grain boundary signal. This can be performed using a "watershed" algorithm, a technique commonly used in image processing for segmentation (Vincent and Soille, 1991). In this approach the grayscale of the image is regarded as a topographic map, with the elevation being represented by the pixel intensity. The task then





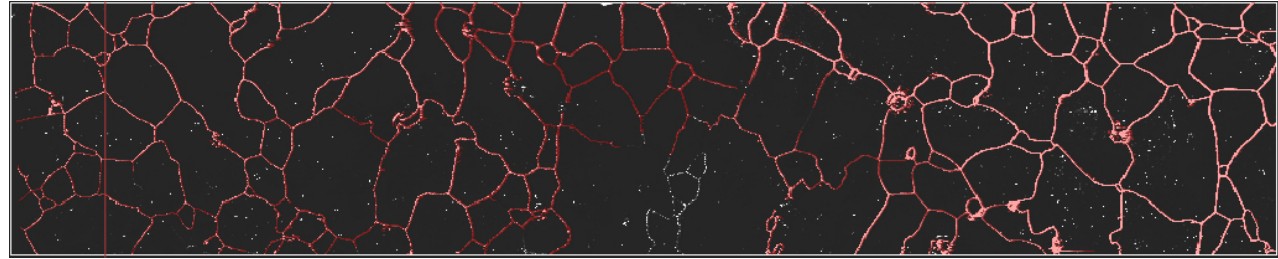

**Figure 6.** Exemplary results of the watershed algorithm used in HDIP for Na map segmentation of the MIS2 sample. The red mark-up is used to classify pixels belonging to grain boundaries. The complement is associated with grain interiors. Note how a small portion in the lower center part was missed by the semi-automated procedure due to a lack of coherency in the Na-grain boundary association. The vertical red stripe on the left image side is an artifact from figure creation.

is to find the "crest lines" in the topographic map. This analysis was performed semi-automatically using a watershed-type
algorithm within the software HDIP: Bright pixels in the grain boundaries are selected manually as a starting point, and the flood tolerance parameter is increased iteratively until the connecting network lines are selected. This procedure is repeated until the grain boundaries of the image have been selected as a "region of interest". The result is two sub-sets of pixels in the image: Pixels associated either with a grain boundary or grain interior (Figure 6). Then, basic statistics are performed on the two subsets individually. Table 3 summarizes the results for the MIS 2 and MIS 5.5 samples.

Based on the coarse assumption that ablation differences between the glass reference materials and the ice samples can be neglected, the ratios of intensities can be converted into elemental ratios using NIST 612 as a reference (Longerich et al., 1996; Jochum et al., 2011). Following this previously employed approach (Della Lunga et al., 2014), Na/Mg, Na/Sr and Mg/Sr are converted accordingly in Table 3.

The ratios reveal that the relative enrichment at grain boundaries is generally highest for Na, between 3-6 times higher as for
Mg and around 10 time higher than for Sr. Next, the relative enrichment at grain boundaries is 3-5 times higher in MIS 5.5 compared to MIS 2. Both effects translate into an analogue situation for the ratios, with the exception of the Mg/Sr ratio. In grain boundaries, the latter shows only comparatively a small difference between MIS 2 and MIS 5.5.

### 3.4 Spatial significance of single line profiles

In order to simulate how the spatial impurity distribution would appear in coarser resolution LA-ICP-MS elemental imaging,
the 35 $\mu$m resolution images are sub-sampled in longitudinal (along scan direction) and transversal direction. The latter is primarily simulating using a larger spot size whereas the decrease in longitudinal direction additionally corresponds to longer washout times. The rows of the original images are averaged stepwise in increments of 1 line, making the transversal resolution decrease in 35 $\mu$m steps. To decrease the longitudinal resolution by approximately the same step, gaussian filtering is applied subsequently to each line with a kernel size adjusted accordingly. Using a gaussian filter along the scan direction in each
line mimics the effects of washout and the moving laser. This is not needed in the transversal direction since individual lines



**Table 3.** Results from basic statistics after image segmentation into grain boundaries and grain interiors. Note that Na, Mg and Sr are intensities in counts per second, whereas Na/Mg, Na/Sr and Mg/Sr are reported as elemental ratios (see text).

| | EDC MIS 2 | | | EDC MIS 5.5 | | |
| | I Boundaries | II Interiors | I/II | I Boundaries | II Interiors | I/II |
| | Mean $\pm 1\sigma$ | Mean $\pm 1\sigma$ | | Mean $\pm 1\sigma$ | Mean $\pm 1\sigma$ | |
|---|---|---|---|---|---|---|
| Na | $(2.00 \pm 0.03)\ 10^6$ | $(4.13 \pm 0.02)\ 10^4$ | 49 | $(1.50 \pm 0.02)\ 10^7$ | $(8.2 \pm 0.1)\ 10^4$ | 183 |
| Mg | $3557 \pm 51$ | $450 \pm 4$ | 8 | $11687 \pm 243$ | $233 \pm 3$ | 50 |
| Sr | $685 \pm 12$ | $179 \pm 2$ | 4 | $3635 \pm 83$ | $196 \pm 3$ | 19 |
| Na/Mg | $200 \pm 4$ | $9.0 \pm 0.1$ | 22 | $471 \pm 12$ | $8.2 \pm 0.2$ | 57 |
| Na/Sr | $11446 \pm 213$ | $327 \pm 2$ | 35 | $37521 \pm 837$ | $431 \pm 9$ | 87 |
| Mg/Sr | $79 \pm 1$ | $19.1 \pm 0.1$ | 4 | $128 \pm 3$ | $11.8 \pm 0.1$ | 11 |

are essentially independent samples. It is important to note that this analysis assumes the continued presence of optimized instrumental settings, thus no further artifacts are introduced. In order to assess the spatial significance of a single longitudinal line with respect to the transversal dimension, the correlation matrix (using the PCC) between all lines in the image is calculated for each step. In the ideal case, all lines in the image look the same, thus making the vertical positioning of the line on the sample surface irrelevant. This would result in fully correlated lines and a correlation matrix filled with ones. To quantify the degree of inhomogeneity of the correlation matrix, the relative standard deviation (RSD) of the entries is reported.

Results are shown here for the case of Na, since featuring the strongest imprint of the grain boundaries. Results for Mg are in the Supplementary Material, being almost identical to the results for Sr (hence not shown). Figures 7, 8 and 9 show the results for the TD Holocene, EDC MIS 2 and EDC MIS 5.5 image, respectively. It becomes evident that the signal of the small-scale bright spots has vanished at 210 $\mu$m, while the majority of the grain boundary remains visible and traceable throughout the images. The presence of the spatially coherent signal corresponding to the grain boundaries is related to the grain size: Large grains remain visible even at 420 $\mu$m, while comparatively smaller grain cannot be distinguished anymore. This becomes especially clear when comparing the 420 $\mu$m images of EDC MIS 2 and EDC MIS 5.5, Figure 8 and 9, respectively. At a scale of 700 $\mu$m, the TD Holocene and EDC MIS 2 images resemble mostly the large-scale intensity gradients. At this point, a high degree of spatial significance of a single line is achieved, meaning that there is only little influence by the relative transversal position of the scan line. Notably, this situation is different for the EDC MIS 5.5 images, comprised by comparatively large grains. Regarding Mg, a comparable degree of homogeneity as for Na is achieved at the steps shown here, indicated by similar relative standard deviation (RSD) values (Supplementary Material).





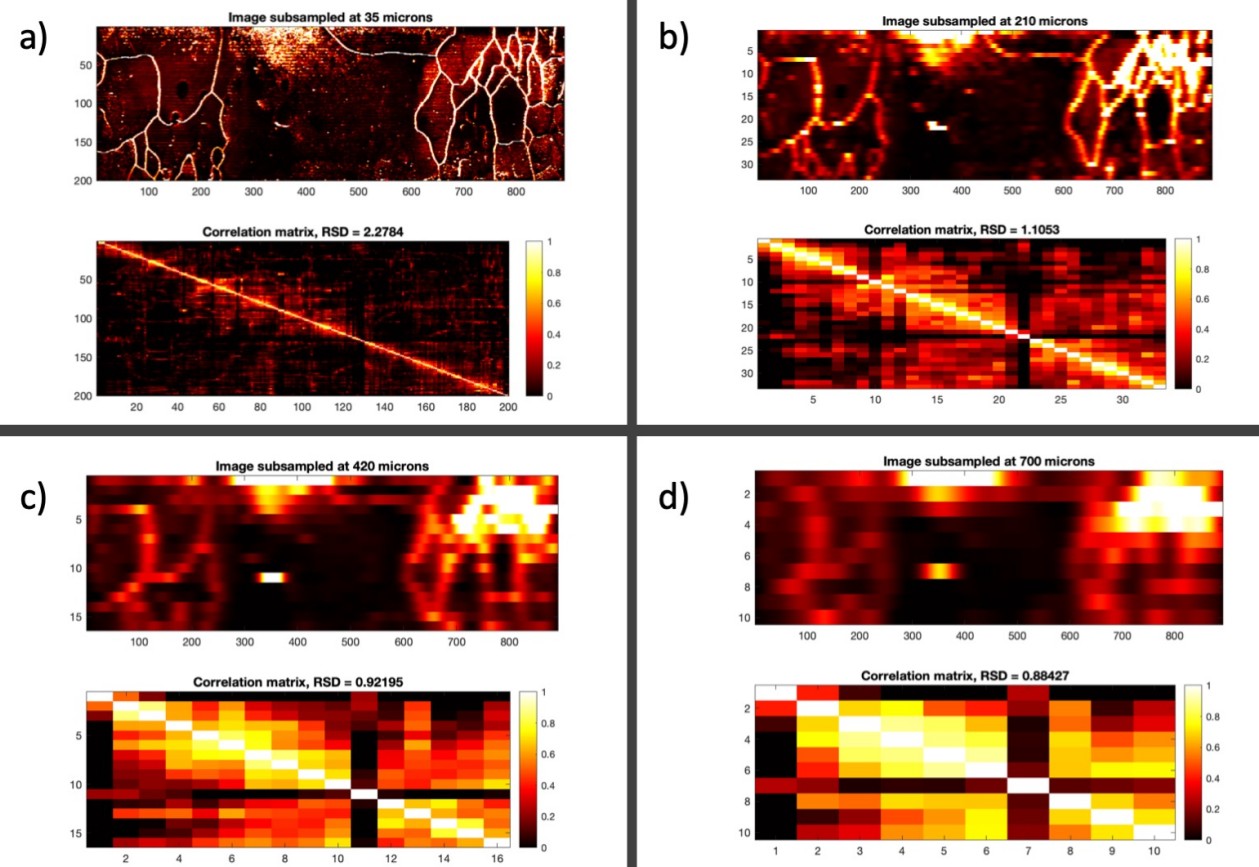

**Figure 7.** Exemplary images illustrating decreasing the spatial resolution of the original image (a) in 35 $\mu$m steps in vertical and horizontal direction (see text). The correlation matrix is calculated from all lines in the sub-sampled images, together with its relative standard deviation (RSD). Shown here are results for the TD Holocene Na image, at steps of 210, 420 and 700 $\mu$m, in tile (b), (c), (d), respectively.

## 4 Discussion

The results are discussed in view of the two-fold exploration into the use of the LA-ICP-MS imaging technique to reveal i) 2D maps of the impurity distribution in relation to the grain boundary network and ii) constraints arising for the interpretation of line profiles as high-resolution stratigraphic signals, allowing to adapt the experimental design accordingly.

### 4.1 Impurity localization and grain boundary network

The two EDC images not only corroborate the previously observed correlation between high Na intensity and locations of
grain boundaries in the TD Holocene sample (Bohleber et al., 2020) but extent the view to samples from core section with very different physical and chemical properties. The three maps exhibit clear differences in their general features and the degree of intra-grain versus in-grain elemental concentration, the latter generally increasing from Holocene – MIS2 – MIS 5.5 for all



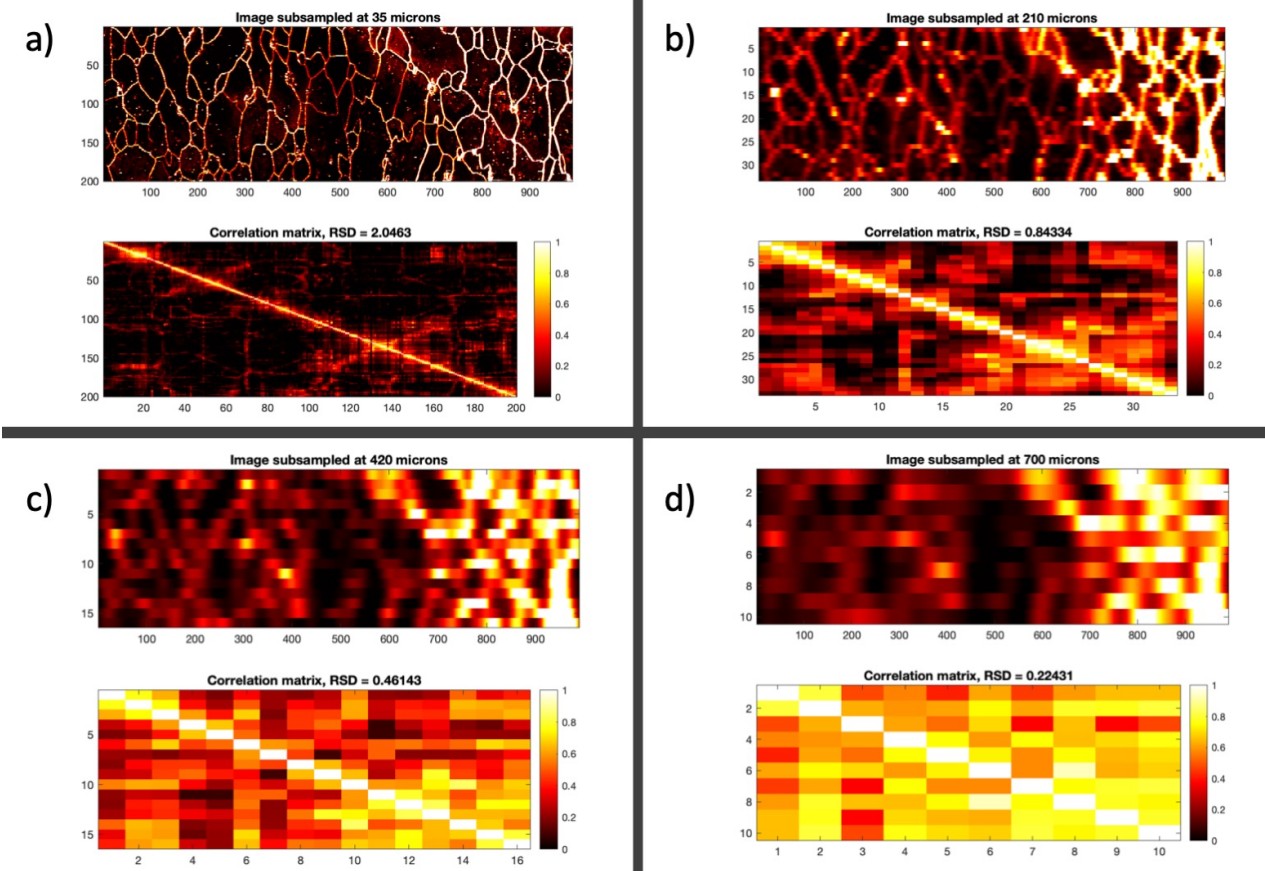

**Figure 8.** Same as Figure 7 but for the EDC MIS 2 image.

elements. This difference argues against sample contamination or surface changes playing a dominant role for the 2D impurity maps, which arguably would make all samples look similar. The already demonstrated reproducibility of the images does

also not support such artifacts (Bohleber et al., 2020). The established rigorous decontamination methods have been proven adequate for LA-ICP-MS ice core analysis (Della Lunga et al., 2014, 2017). Notably, the general tendency of Na and, to a lesser extent, Mg to localize at grain boundaries is consistent with LA-ICP-MS impurity mapping analyzing ice samples of Greenland ice cores. However, prior to the imaging technique, elemental maps had to be acquired using arrays of laser spots with spot sizes larger than 100 $\mu$m, followed by spatial interpolation. Consequently, this method faced limitations regarding

investigations on impurity localization at grain boundaries (Della Lunga et al., 2014, 2017). In this context, the artifact-free images generated at 35 $\mu$m resolution and without further interpolation provide a significant step forward.

Considering the comparison of the two EDC samples from MIS 2 and MIS 5.5 in more detail, the in-grain signals, distributed intensity and individual bright spots, stand out in the MIS 2 maps but are almost entirely absent in MIS 5.5. Assuming a relation between bright spots and micro-inclusions or dust-related particles, this difference in map features between the glacial



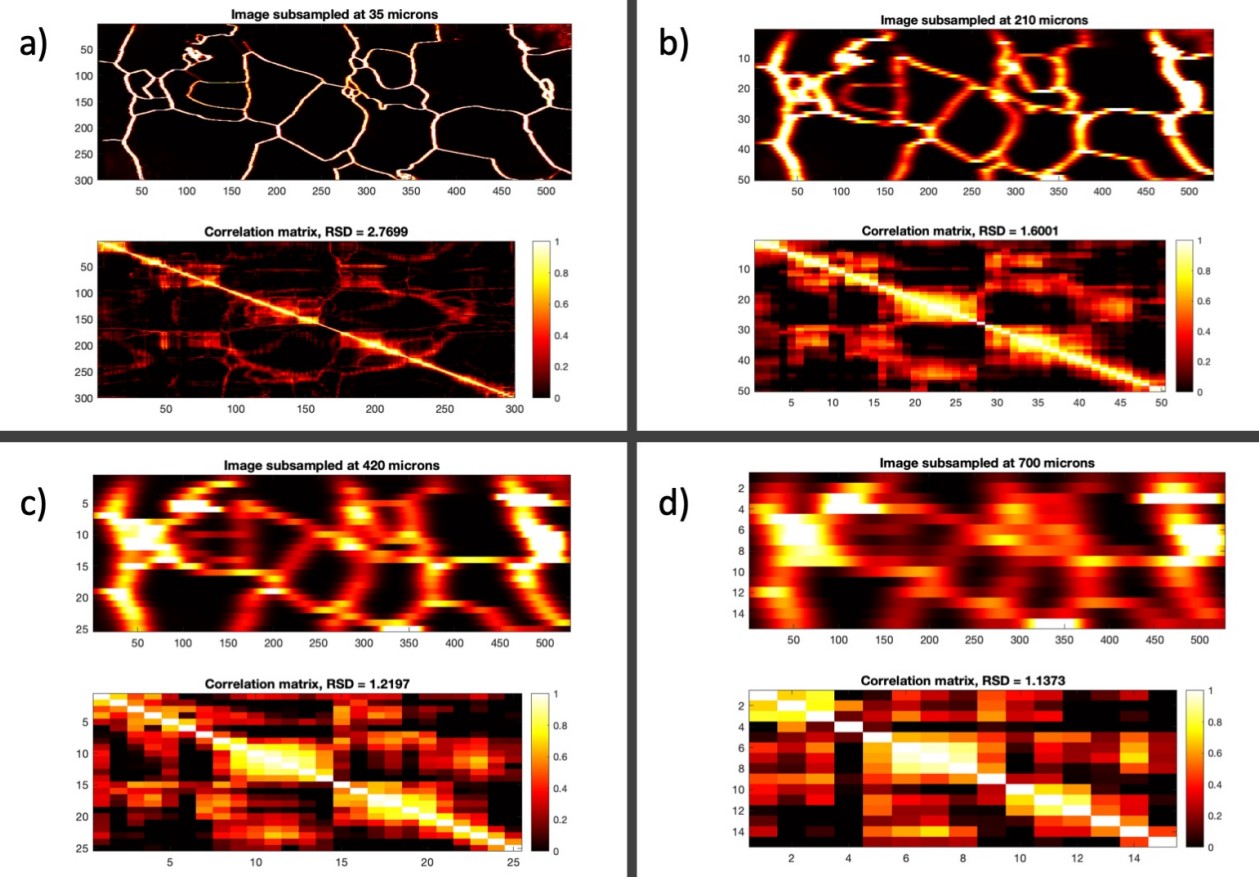

**Figure 9.** Same as Figure 7 but for the EDC MIS 5.5 image.

and interglacial sample is consistent with the macroscopic chemistry: glacial impurity (including dust) concentrations are generally up to several orders of magnitude higher than the interglacial ones (e.g. Fischer et al., 2007; Lambert et al., 2008). The known smaller average crystal area in glacial samples is also observed, especially through the Na map. In the MIS 5.5 sample also Mg and Sr are located predominantly at grain boundaries, with only minor signals from the grain interiors.

Based on the indicators used for co-localization analysis in the LA-ICP-MS images, Mg does not show a clear preference for

neither Na (related mostly to sea-salt) nor Sr (a tentative substitute for terrestrial dust sources more commonly investigated through Ca). Notably, Mg may have fractions related to sea salt as well as terrestrial dust. By comparison to present conditions, the last glacial period stands out by a major contribution from insoluble components and, to a lesser degree, terrestrial salts (Legrand and Mayewski, 1997).

In the framework of the observed differences in the maps of MIS 2 and MIS 5.5, a simplified hypothesis is that, depending on

their individual mobilities, the comparatively mobile species have migrated to the grain boundaries whereas less mobile species such as the insoluble particulate fraction (terrestrial dust) remains preferably within the grains. This is a simplified view because





particulate inclusion may show "pinning of" or "dragging with" grain boundaries, causing their localization also at boundaries (Faria et al., 2014b). However, this view raises the question if differences in chemical composition can be observed in different regions of the image, and if this may indicate a different degree of localization for soluble and insoluble fractions. It is worth
noting in this context that LA-ICP-MS measures the total impurity amount, and thus cannot directly distinguish soluble and insoluble fractions. A more detailed analysis of the glacio-chemistry would clearly benefit from measurement of additional elements that can serve as references for the respective fractions.

However, the high-resolution images create additional alternative opportunities to overcome such limitations by introducing image analysis techniques. This approach was explored here to a limited degree, to exemplarily compare intra-grain vs. in-grain
signals. It is worth pointing out that this type of analysis using image segmentation was performed as a post-processing step and did not require a separate experimental design. Experiments aimed at comparison of intra-grain vs. in-grain were previously performed with LA-ICP-MS but required the manual tracking of the grain boundaries with the laser scan (Beers et al., 2020; Kerch, 2016).

It should also be noted again that imaging the localization of impurities does not require a fully quantitative method. However,
the ratio of intensities, either between boundaries and interiors, or among two elemental species, are robust indicators of the absolute distribution also without calibration. The fact that the enrichment at grain boundaries is generally highest for Na, followed by Mg and Sr, suggests that on the micron-scale, distinct differences in the interaction with the grain boundary network exist among these analytes and among ice from different climatic periods: The results indicate that in the comparatively warmer and less dusty conditions of MIS 5.5, a larger fraction of the here analyzed impurities is located at the grain boundaries.
This is of relevance also regarding the conditions in deep and warm ice: the presence of the impurities at grain boundaries is a prerequisite for their potential migration along the grain boundary network (Rempel et al., 2001). The imaging of Mg could provide additional insight on the already investigate anomalous signals occurring in the EDC chemical records below 2800 m, in particular when combined with the analysis of S (or alike chemical substitutes for sulfate) (Traversi et al., 2009). It is evident that only limited generalized conclusions can be drawn from the small-sized images. Accordingly, it is not intended to
answer the above hypothesis on different behavior of chemical impurities in relation to their mobility and insoluble fractions. However, it becomes clear that the new LA-ICP-MS imaging technique can offer important insights into the ice stratigraphy on the micron-scale and that special merit comes from introducing techniques of image analysis to investigate the chemical images. Future efforts in combining techniques image analysis in an automated way and for even larger images seem highly intriguing in this context.
In the future, a meaningful comparison with elemental concentrations and ratios from other macroscopic techniques, such as continuous flow analysis, calls for larger, cm-sized areas. Regarding an inter-method comparison over smaller areas, the comparison with cryo-Raman analysis or synchrotron light techniques (Baccolo et al., 2018) may provide added value for the investigation of impurity localization, in particular micro-inclusions and dust. In contrast to the findings of the present work, cryo-Raman performed on the EPICA DML core for MIS 6 and MIS 5e detected no signal of significant relationship between
micro-inclusions and grain boundaries, which lacked also a signal of dissolved impurities (Eichler et al., 2019). Although major





challenges may arise due to methodological differences, a direct comparison between LA-ICP-MS and these two techniques seems a worthwhile future task.

## 4.2 Spatial significance of line profiles and future experimental designs

In addition to the investigation of impurity localization, LA-ICP-MS ice core analysis has received special interest for detecting
highly thinned stratigraphic layers as high-resolution paleoclimate records. For ice cores from coastal Antarctica, the resolution of a few hundred microns can offer the detection of layers to the annual timescale even in deeper layers (Haines et al., 2016). Following studies utilizing the comparison with continuous flow analysis, further smoothing is commonly applied to the LA-ICP-MS signals prior to such interpretation (Della Lunga et al., 2017; Bohleber et al., 2018).

At the resolution of 35 $\mu$m, the present results demonstrate that crystal features such as grain boundaries determine the high-
frequency signal components in single line profiles. This means that signals obtained from line profiles will be greatly influenced by their positioning on the surface, i.e. transversal to the core axis and scan direction. In contrast to this, the central hypothesis is that the original stratigraphy resulting from paleoclimatic variability should produce signals that are not a function of the transversal position of the scan line. Measuring multiple parallel scan lines should show a high degree of shared signal and thus have a high spatial significance. Notably, this ideal case may already be flawed once so-called micro-folds
(Svensson et al., 2005) start to appear.

This raises the following problem: on the one hand, for the investigation of paleoclimatic signals on a mm- to $\mu$m-scale, the measurement of single line profiles across the main core axis is preferred in order to avoid the comparatively time- and resource-intensive nature of the imaging technique. On the other, only the latter can provide the required detail regarding the signal imprints arising from ice crystal features such as grain boundaries. This means that an experimental design based on
imaging should first set the choice of spatial scale (measurement resolution), so that the spatial coherence is maximized and the grain boundary imprint minimized.

The results from simulating coarser resolution images show how this can be achieved based on the high-resolution images. For the TD Holocene and EDC MIS 2 image a resolution of $400 - 700$ $\mu$m can be sufficient to achieve transversal signal coherence. This is consistent with the need to further reduce the resolution in previous studies using spot sizes around 200 $\mu$m (Sneed
et al., 2015; Della Lunga et al., 2017; Spaulding et al., 2017; Bohleber et al., 2018). The instance of the EDC MIS 5.5 image shows that this range is not a generally applicable value, however. Due to the larger grains the signals remain substantially heterogenous in the transversal direction.

It is clear that maximizing spatial coherence will immediately depend on: i) the degree of localization of an impurity species at the grain boundaries, and ii) the average size of the grains with respect to the stratigraphic layering of interest. Both can be
taken into account with the 2D impurity imaging technique. Considering an average grain radius around $5 - 6$ mm in the deeper sections of the EDC ice core (EPICA Community Members, 2004), the required size of the maps should be substantially larger than the present ones, however.

The increase in map size is a considerable, yet solvable technical and practical challenge. Even with the high scan speed employed here, the recording of a 7 x 10.5 mm image requires 200 parallel scan lines at a 35 mm spot size, corresponding to 1.8



million individual laser shots fired. With unidirectional measurements the recording takes around 2.5 hours, although imaging time can be reduced by about 30% if scanning in bidirectional mode. The latest technological developments in LA-ICP-MS imaging promise further advancement in speed through faster washout and higher repetition rate lasers (Šala et al., 2020). Regarding stratigraphic line scans with LA-ICP-MS, the already developed large cryogenic chambers can eliminate the need for preparing cm-sized samples (Sneed et al., 2015). If combined with the scan speed achieved in the present method, the line

scan of a 55 cm core section would be completed in under 10 mins. In this framework, the present study clearly demonstrates the merit of the LA-ICP-MS imaging technique both regarding studying impurity localization and setting the experimental design for stratigraphic investigations. The imaging approach should hence be integrated in future efforts in ice core analysis with LA-ICP-MS.

## 5 Conclusions

Through the integration of state-of-the-art imaging techniques, LA-ICP-MS ice core analysis has taken the step from 1D into 2D. The next step in LA-ICP-MS ice core analysis now offers scan speeds increased by one order of magnitude for single line profiles and the ability to map the localization of impurities at high spatial resolution (35 $\mu$m). The present work has demonstrated the new potential for investigating the location of impurities and to improve the interpretation of single line profiles considering imprints from ice crystal features. Two-dimensional chemical imaging with LA-ICP-MS showed

distinct differences among glacial and interglacial samples of the Talos Dome and EPICA Dome C ice cores from central Antarctica. The images reveal that grain boundaries coincide with high intensities of Na for all samples. Mg and Sr are more distributed in the Talos Dome Holocene sample and the glacial sample from EPICA Dome C. The interglacial sample from MIS 5.5 shows all elements predominantly at grain boundaries. This finding is corroborated by introducing image segmentation techniques to separately quantify in-grain vs. intra-grain intensities as well as their ratios. Simulation of reduced measurement

resolution shows that the spatial significance of a single line profile increases as the imprint of grain-boundaries weakens for coarser resolution. This allows to adapt settings specifically fit-for-purpose, e.g.to avoid misinterpretation of ultra-fine resolution signals in the presence of ice crystal imprints. An immediate future target is imaging over larger areas to increase their spatial significance, in particular for investigations in deep ice with cm-sized grains. In this regard, the present findings have clearly demonstrated the merit of driving forward the LA-ICP-MS ice core imaging technique.

*Data availability.* The underlying data will be made available via a public data repository (e.g. Pangaea) after completion of the peer-review process.

*Author contributions.* PB conducted the measurements with help of MR. The experimental design was developed by PB with the help of MR, MS and CB. PB wrote an initial version of the manuscript. All authors contributed to the discussion of the results and the final version of the manuscript.



*Competing interests.* The authors declared that there are no competing interests.

*Acknowledgements.* The authors thank Ciprian Stremtan and Stijn van Malderen for their technical support. Likewise we thank Alessandro Bonetto for support in the laboratory and Luca Fiorini for assistance with the spatial significance experiments. Marcello Pelillo and Sebastiano Vascon are gratefully acknowledged for helpful discussions on image analysis. PB gratefully acknowledges funding from the European Union's Horizon 2020 research and innovation programme under the Marie Skłodowska-Curie grant agreement No. 790280. MS
acknowledges support from the Slovenian Research Agency (ARRS), contract number P1-0034. ELGA LabWater is acknowledged for providing the PURELAB Option-Q and Ultra Analytic systems which produced the ultrapure water used for cleaning and decontamination. This publication was generated in the frame of Beyond EPICA. The project has received funding from the European Union's Horizon 2020 research and innovation programme under grant agreement No. 815384 (Oldest Ice Core). It is supported by national partners and funding agencies in Belgium, Denmark, France, Germany, Italy, Norway, Sweden, Switzerland, The Netherlands and the United Kingdom. Logistic
support is mainly provided by PNRA and IPEV through the Concordia Station system. The opinions expressed and arguments employed herein do not necessarily reflect the official views of the European Union funding agency or other national funding bodies. This is Beyond EPICA publication number xy. The Talos Dome Ice Core Project (TALDICE), a joint European programme led by Italy, is funded by national contributions from Italy, France, Germany, Switzerland and the United Kingdom. This is TALDICE publication n xy.



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
