# Peer review of "Two-dimensional impurity imaging in deep Antarctic ice cores: Snapshots of three climatic periods and implications for high-resolution signal interpretation"

_The Cryosphere, 2020_

## Referee Comment (RC1) · David M. Chew (Referee) · 7 Feb 2021

Dear editor,

This is an interesting study employing LA-ICP-MS mapping of ice cores from Antarctica. The glaciology/climatology aspects are not my area of expertise, so my substantive comments below mainly concern the methodology. The paper is generally easy to follow, but there are many instances of awkward phrasing. I have a list of suggested typographical improvements below, but the paper should have a quick edit by a native

[Figure]

English speaker. I recommend minor revisions.

A washout of 34 ms is quoted (i.e. the system is capable of returning to baseline with a repletion rate of 29Hz). Yet it says in the paper L70-71 "With washout times in the tens of ms range, the recording of baseline-separated single pulses at high repetition rates becomes possible; 294 Hz and a dosage of 10 were used here". There is no way with a washout of 34 ms that that you would see baseline-separated single pulses, so some rewording is needed here. Additionally, the term "dosage" is not used all that commonly in the LA-ICP-MS literature. I would define it in one sentence, and the recent JAAS article by Šala et al. could be cited.

The isotopes 23Na, 25Mg and 88Sr were measured, with dwell times of 4, 4.6 and 10 ms respectively. What was the total sweep time (i.e. including settling) and the duty cycle?

L138-140 "The relative higher background level seen in Na has been observed before in LA-ICP-MS ice core analysis and was suggested to be related to the use of NIST glasses as reference materials (Della Lunga et al., 2017)." Same would probably apply to any soda-lime glass. But my main query here were the signal intensity maps not background-corrected? And if not, why?

Typographical improvements L54 "In presence of a variable signal" – reword start of sentence.

L56 delete "on this ground"

L63 "keeps the ice samples surface temperature durably at" – change to "keeps the surface temperature of the ice samples consistently at"

L91 "Sample selection was guided to consider ice of" change to "Sample selection targeted ice at"

L93 change to "calls for mapping large areas"

L99 change to "local maximum in grain radius at around 3.5 mm"

L106 use of "sections" is confusing in this sentence. Are we talking about different samples, or area / domains within a sample.

L109 delete 'their'

L121 "In-grain intensities of Mg and Sr" is not clear.

L129 change to "in the Mg and Sr signal distribution"

L133 delete "the image of"

L146 change to "since they are superior in such cases"

L159-160 change to "allows image segmentation based solely on the LA-ICP-MS images to be performed"

L174 change to "between 3-6 times higher than for"

L176 and 177. I do not follow either of these two sentences" "Both effects translate into an analogue situation for the ratios, with the exception of the Mg/Sr ratio. In grain boundaries, the latter shows only comparatively a small difference between MIS 2 and MIS 5.5."

L186-7 delete "It is important to note that this analysis assumes the continued presence of optimized instrumental settings, thus no further artifacts are introduced."

L188 what is the "transversal dimension"? Do not follow.

L192 change to "since it features"

L197 change to "while comparatively smaller grains"

L200 change to "only a small influence". I do not follow "the relative transversal position" part of the sentence.

L202 delete "at the steps shown here"

[Figure]

L210 change to "but extend approach to samples from core sections"

L217 replace "analyzing" with "of"

L218-9 reword to "However, prior to the advent of the LA-ICP-MS imaging technique, elemental maps had to be acquired using arrays (grids) of laser spots with spot sizes larger than 100 $\mu$m, followed by spatial interpolation"

L231 change "may have fractions" to "may be"

L237 I do not follow 'may show "pinning of" or "dragging with"

L244 delete "exemplarily" (this word is used incorrectly in all instances in the paper

L254 delete "here analyzed"

L257 delete "already investigate"

L262-3 "image analysis applied to investigating the chemical images is advantageous"

L269 delete "signal of"

L272 replace "task" with "goal"

L296 change to "not a generally applicable value, however as the larger grains"

L305 change "recording" to "imaging"

L311 change "regarding" to "for"

L321-2 "are more distributed" is not clear

L324-6 change to "Simulations of coarser resolution experiments shows that the spatial significance of a single line profile increases as the imprint of grain-boundaries weakens at coarser resolution."

L326 change to "This allows settings to be adapted specifically fit-for-purpose"

Figure 5 caption. Change second sentence to "A linear regression (red dashed line) is
shown purely as a visual aid."

Figure 7 caption. Change first sentence to "Example images illustrating the effect of decreasing the spatial resolution of the original image (a) in 35 $\mu$m steps in the vertical and horizontal direction (see text).

Table 2 caption. Delete "Overview on results from"

---

## Referee Comment (RC2) · Anonymous Referee #2 · 10 May 2021

This paper shows some of the first images produced with a new laser ablation – ICPMS system that is configured to produce two dimensional maps at high resolution. The paper shows maps from 3 cores, representing the Holocene at Talos Dome, MIS2 and MIS 5.5 at dome C. In fact the method itself and the results from the Talos Dome Holocene core have already been presented (in the authors' JAAS paper). However this paper is definitely an advance in that it shows the wide applicability and potential of the method, displays some beautiful images for the glaciological community, and

considers some issues related to how such a method should be used, processed and imaged.

The highlight of the paper is certainly the lovely images we see in Figs 2-4. These really area fine technical achievement and a joy to look at and think about. The paper considers the differences between elements (Na, Mg and Sr), and the differences between climate periods. The second of these is indicative but difficult to pursue: with only one example from each climate period, can we be sure that the findings are typical? I accept that it is unreasonable to expect more at this stage, and I am willing to ignore this problem this time. However in the future it will be necessary to see enough different sections in each climate period to really understand the rules.

The discussion of how to average the records in order to use the method to its best effect is important, but is not very well-explained. I think I got it in the end, and the result is worth discussing, but I will suggest some better explanation of what was done. I like the thinking in this section though – until now it seems to have been assumed that better resolution is always good. Here the authors show clearly that better resolution helps with understanding microstructure, but will have to be sacrificed to understand large-scale layering.

Overall, I do see some ways in which the explanations in the paper could be improved. But as a well-illustrated proof of concept this is an excellent paper and should be published.

Detailed comments:

Page 1, line 11 "it is demonstrated how instrumental settings can be adapted specifically fit-for-purpose". This doesn't quite make sense, I suggest "it is demonstrated how instrumental settings can be adapted to be fit-for-purpose".

Line 41. I suspect this became available after the paper was prepared but the authors may wish to reference Ng et al 2021 here as well as Rempel et al.

Line 68-71. Like the other reviewer, I didn't understand how one could reach 294 Hz if the washout time is 34 ms. Please explain this further.

Line 75. I don't think you mean 150 mm square! Maybe 150 um? But anyway please be clear whether this means 150 x 150, rather than a size that amounts to an area of 150 umˆ2.

Figs 2-4. I really like the elemental maps but am a little less clear what I am seeing in the composites in part c. Perhaps it's just the colour scale that is confusing me, because superimposing even the lightest colours shown there will certainly not give a white. Should the scales run through to very light blue/red/green to more correctly characterise what you did?

Line 140. I don't really understand this discussion which leads to the discussion about the use of NIST glass reference standards. I can understand that the instrument can be more sensitive to Na, and that Na is at higher concentrations so should give higher counts. But I'm not understanding how the standards would affect the background or why this is relevant. Do you mean that there is a contamination background because of the standard? But then you're clearly seeing a stronger signal response as well as a background response for Na. As you can see I am confused so please explain what you are suggesting here.

Line 170 and Table 3. Are the elemental ratios in Table 3 ratios by weight or molar ratios?

Around lines 170 and 230: You seem to suggest maybe the marine material is at the grain boundaries and the crustal dust in the grains. While this makes sense the ratio of Na/Mg in the grain boundaries is much higher than that of sea salt. Might be worth discussion.

Page 11. I found it really hard to follow what the correlation matrices in Figs 7-9 are. I think I got it in the end but please spell it out. If I have understood correctly you have

taken all the parallel vertical profiles (ie at 420 um resolution you'd have 10 parallel profiles) and correlated them against all the others. This should then lead to a symmetrical pattern where perfect correlations would be white across the entire diagram. Please explain it in these kind of simple terms. I think it's harder to grasp because you have put the figures as rectangles rather than squares, leading the reader to think they might be looking at a map, and also to the plot not looking symmetrical.

---

## Author Comment (AC1) · 2 Jun 2021

**Manuscript TC-2020-369**

**"**Two-dimensional impurity imaging in deep Antarctic ice cores: Snapshots of three climatic periods and implications for high-resolution signal interpretation**"**

by Pascal Bohleber, Marco Roman, Martin Šala, Barbara Delmonte, Barbara Stenni and

Carlo Barbante

- Response to reviews -

***Please note:***

- ***All line numbers in "Changes to manuscript" refer to the new revised version***

***(if not noted otherwise)***

- ***Changes in the revised pdf are highlighted in red***

- ***Author's responses to the referee's comments are in blue***

**Overview on manuscript revision**

We thank both referees for their positive and helpful reviews of our manuscript. The revision comprised the following main changes:

- The presentation of the imaging method was clarified regarding the connection between fast washout and high repetition rate (Section 2).

- The assessment of the spatial significance of line profiles (Section 3.4) was clarified in more simple terms to improve readability.

- The discussion on impurity localization was re-organized to better separate the discussion of the chemical images and aspects regarding the imaging method (Section 4.1).

- Figure 7,8,9 were changed to include the correlation matrix as a square plot. The

Figures in the Supplementary Material were changed accordingly.

We believe that these changes have substantially improved the manuscript. The responses to the specific comments and technical corrections are detailed below (in blue) together with the track changes in the original manuscript (in red) which is at the end of this document.

**Response to referee #1 David M. Chew**

Dear editor,

This is an interesting study employing LA-ICP-MS mapping of ice cores from Antarctica. The glaciology/climatology aspects are not my area of expertise, so my substantive comments below mainly concern the methodology. The paper is generally easy to follow, but there are many instances of awkward phrasing. I have a list of suggested typographical improvements below, but the paper should have a quick edit by a native English speaker. I recommend minor revisions.

We thank the referee for the comments, which especially helped to present the methodology more clearly. We have addressed all comments as described below and have also tried to improve the readability of the text with the help of a native English speaker.

A washout of 34 ms is quoted (i.e. the system is capable of returning to baseline with a repletion rate of 29Hz). Yet it says in the paper L70-71 "With washout times in the tens of ms range, the recording of baseline-separated single pulses at high repetition rates becomes possible; 294 Hz and a dosage of 10 were used here". There is no way with a washout of 34 ms that that you would see baseline-separated single pulses, so some rewording is needed here. Additionally, the term "dosage" is not used all that commonly in the LA-ICP-MS literature. I would define it in one sentence, and the recent JAAS article by Šala et al. could be cited.

We now realize that the two sentences can be misunderstood. With a dosage of 10 we improve the image quality but do not separate individual pulses anymore. To avoid this misunderstanding, we decided to separate the general statement regarding the benefit of fast washout and the specific statement regarding our acquisition settings. The general statement is now moved to the introduction, where the use of fast washout technology was already mentioned (line 31). This way, we are focusing in the method section solely on the description of our acquisition settings. We are also including the suggested reference by Šala et al. and give an explicit explanation of the term "dosage" (line 74).

employing dedicated ablation cells with fast washout as well as optimizing the lasing and ICP-MS settings have introduced a new state-of-the-art in imaging techniques with LA-ICP-MS (Wang et al., 2013; van Elteren et al., 2019). The term "washout time" refers to the time needed to transfer the ablated sample aerosol plume to the ICP-MS. It is principally determined by the extraction efficiency from the ablation cell and any subsequent dispersion in the transfer line. With washout times in the tens of ms range, the recording of baseline-separated single pulses at high repetition rates becomes possible (Van Malderen et al.,

2015). Recently, this new imaging approach was transferred to ice core analysis with LA-ICP-MS, offering the opportunity

> ARIS, a rapid aerosol transfer line, was used, resulting in a washout times of ~34 ms. A repetition rate of 294 Hz and a dosage
> 75 of 10 was used here. In contrast to single pulse analysis, a dosage greater than 1 implies that each pixel is generated by multiple
> partially overlapping laser shots, which leads to an improved signal-to-noise ratio and better image quality (Šala et al., 2021).
> The fast washout combined with a high repetition rate allows scanning of the surface at around one millimeter per second,
> which is roughly 10 times faster than previous studies on ice cores (Della Lunga et al., 2017; Spaulding et al., 2017). As a

The isotopes 23Na, 25Mg and 88Sr were measured, with dwell times of 4, 4.6 and 10 ms respectively. What was the total sweep time (i.e. including settling) and the duty cycle?

The total sweep time was set to 34 ms, matching the washout time in order to avoid image artefacts. We routinely acquired four analytes, including Na, Mg, Sr and the additional mass 55Mn, the latter with a dwell time of 10 ms. This results into a total duty cycle of ~84%. We added this information to the text. (line 86).

> 85 precise synchronization of data acquisition required to avoid image artifacts, the number of analytes/isotopes was restricted.
> Four elements were routinely recorded per image: $^{23}$Na, $^{25}$Mg, $^{55}$Mn and $^{88}$Sr with respective ICP-MS dwell times of 4, 4.6,
> 10 and 10 ms. (Bohleber et al., 2020). The total sweep time was 34 ms, specifically set to match the washout time, resulting in
> a total duty cycle of 84%. Considered in the following are Na, Mg and Sr, due to their significance as paleoclimate proxies in
> polar ice cores (Legrand and Mayewski, 1997): Na being related mostly to sea-salt, Mg with both marine and terrestrial sources

L138-140 "The relative higher background level seen in Na has been observed before in LA-ICP-MS ice core analysis and was suggested to be related to the use of NIST glasses as reference materials (Della Lunga et al., 2017)." Same would probably apply to any soda-lime glass. But my main query here were the signal intensity maps not background-corrected? And if not, why?

Following the referees' comments, we find that we have to clarify here the fact that the higher levels observed for Na are mainly due a higher (absolute) instrumental sensitivity for the element, but we cannot exclude some memory effect due to the contextual ablation of glasses for tuning, drift correction and quantification, as hypothesized by Della Lunga et al. 2017. We decided to reword this paragraph to avoid this potential misunderstanding (line 147). To answer the question: Yes, the signal intensity maps were in fact background and drift corrected, this is already explicitly stated in Lines 84-85 of the original manuscript.

> For further comparison of the degree of co-localization, the matrices of intensity values that underlie the images shown in
> Figure 2, 3 and 4 were used to make scatter plots for each pair of elements. As becomes evident from Figure 5, the intensities
> for Mg and Sr are generally similar, while Na intensities can be higher by several orders of magnitude. This difference can
> be explained by higher Na concentrations paired with a higher (absolute) instrumental sensitivity for the element. The scatter
> 150 plots also indicate the almost absent co-localization in the TD Holocene image, showing signs of mutual exclusions (values

Typographical improvements

All suggested changes were made accordingly.

L54 "In presence of a variable signal" – reword start of sentence.

Changed accordingly. The respective sentence was reworded.

various depth sections were selected, that were representative of distinct climatic periods. The samples were analyzed, aiming to include a broad spectrum of ice properties, such as age and mean grain size. These snapshots of the 2D impurity distribution taken by LA-ICP-MS elemental imaging, provide important details on the location of impurities in relation to the grain boundary network. The imprint of the grain boundaries may vary between different impurity species and climatic periods. Consequently, the spatial significance of a single line profile along the main core axis has to be carefully assessed. These 2D images provide new and improved information for this purpose. It has also been shown how measurement settings can be adapted so LA-ICP-MS line profiles can be used when investigating climate proxy signals in highly thinned deep polar ice.

L56 delete "on this ground"

Changed accordingly.

L63 "keeps the ice samples surface temperature durably at" – change to "keeps the surface temperature of the ice samples consistently at"

Changed accordingly.

L91 "Sample selection was guided to consider ice of" change to "Sample selection targeted ice at"

Changed accordingly.

L93 change to "calls for mapping large areas"

Changed accordingly.

L99 change to "local maximum in grain radius at around 3.5 mm"

Changed accordingly.

L106 use of "sections" is confusing in this sentence. Are we talking about different samples, or area / domains within a sample.

We are actually referring to certain parts of the image. We clarified this sentence accordingly.

[revised manuscript text omitted]

L244 delete "exemplarily" (this word is used incorrectly in all instances in the paper

Changed accordingly (and revised throughout the paper).

L254 delete "here analyzed"

Changed accordingly.

L257 delete "already investigate"

Changed accordingly.

L262-3 "image analysis applied to investigating the chemical images is advantageous"

Changed accordingly.

L269 delete "signal of"

Changed accordingly.

L272 replace "task" with "goal"

Changed accordingly.

L296 change to "not a generally applicable value, however as the larger grains"

Changed accordingly.

L305 change "recording" to "imaging"

Changed accordingly.

L311 change "regarding" to "for"

Changed accordingly.

L321-2 "are more distributed" is not clear

Rephrased to clarify.

differences among glacial and interglacial samples of the Talos Dome and EPICA Dome C ice cores from central Antarctica.

The images reveal that grain boundaries coincide with high intensities of Na for all samples. In the Talos Dome Holocene sample and the glacial sample from EPICA Dome C, Mg and Sr are presented also in the grain interiors. The interglacial

L324-6 change to "Simulations of coarser resolution experiments shows that the spatial significance of a single line profile increases as the imprint of grain-boundaries weakens at coarser resolution."

Changed accordingly.

L326 change to "This allows settings to be adapted specifically fit-for-purpose"

Changed accordingly.

Figure 5 caption. Change second sentence to "A linear regression (red dashed line) is shown purely as a visual aid."

Changed accordingly.

Figure 7 caption. Change first sentence to "Example images illustrating the effect of decreasing the spatial resolution of the original image (a) in 35 μm steps in the vertical and horizontal direction (see text).

Changed accordingly.

Table 2 caption. Delete "Overview on results from"

Changed accordingly.

---

## Author Comment (AC2) · 2 Jun 2021

|                                                                                   | Manuscript TC-2020-369                                                                     |
|-----------------------------------------------------------------------------------|--------------------------------------------------------------------------------------------|
| "Tw                                                                               | o-dimensional impurity imaging in deep Antarctic ice cores: Snapshots of three climatic    |
|                                                                                   | periods and implications for high-resolution signal interpretation"                        |
| by P                                                                              | ascal Bohleber, Marco Roman, Martin Šala, Barbara Delmonte, Barbara Stenni and Carlo       |
|                                                                                   | Barbante                                                                                   |
|                                                                                   | - Response to reviews -                                                                    |
| Please                                                                            | e note:                                                                                    |
| •                                                                                 | All line numbers in "Changes to manuscript" refer to the new revised version (if not       |
|                                                                                   | noted otherwise)                                                                           |
| •                                                                                 | Changes in the revised pdf are highlighted in red                                          |
| •                                                                                 | Author's responses to the referee's comments are in blue                                   |
|                                                                                   |                                                                                            |
|                                                                                   |                                                                                            |
|                                                                                   | Overview on manuscript revision                                                            |
| We tha                                                                            | ank both referees for their positive and helpful reviews of our manuscript. The revision   |
| compr                                                                             | ised the following main changes:                                                           |
| •                                                                                 | The presentation of the imaging method was clarified regarding the connection between      |
|                                                                                   | fast washout and high repetition rate (Section 2).                                         |
| •                                                                                 | The assessment of the spatial significance of line profiles (Section 3.4) was clarified in |
|                                                                                   | more simple terms to improve readability.                                                  |
| ٠                                                                                 | The discussion on impurity localization was re-organized to better separate the            |
|                                                                                   | discussion of the chemical images and aspects regarding the imaging method (Section        |
|                                                                                   | 4.1).                                                                                      |
| •                                                                                 | Figure 7,8,9 were changed to include the correlation matrix as a square plot. The Figures  |
|                                                                                   | in the Supplementary Material were changed accordingly.                                    |
| We bel                                                                            | lieve that these changes have substantially improved the manuscript. The responses to the  |
| specifi                                                                           | c comments and technical corrections are detailed below (in blue) together with the track  |
| changes in the original manuscript (in red) which is at the end of this document. |                                                                                            |

| 29 | Response to anonymous referee #2                                                                            |
|----|-------------------------------------------------------------------------------------------------------------|
| 30 | This paper shows some of the first images produced with a new laser ablation – ICPMS system that is         |
| 31 | configured to produce two dimensional maps at high resolution. The paper shows maps from 3                  |
| 32 | cores, representing the Holocene at Talos Dome, MIS2 and MIS 5.5 at dome C. In fact the method              |
| 33 | itself and the results from the Talos Dome Holocene core have already been presented (in the                |
| 34 | authors' JAAS paper). However this paper is definitely an advance in that it shows the wide                 |
| 35 | applicability and potential of the method, displays some beautiful images for the glaciological             |
| 36 | community, and considers some issues related to how such a method should be used, processed and             |
| 37 | imaged.                                                                                                     |
| 38 | The highlight of the paper is certainly the lovely images we see in Figs 2-4. These really are a fine       |
| 39 | technical achievement and a joy to look at and think about. The paper considers the differences             |
| 40 | between elements (Na, Mg and Sr), and the differences between climate periods. The second of                |
| 41 | these is indicative but difficult to pursue: with only one example from each climate period, can we be      |
| 42 | sure that the findings are typical? I accept that it is unreasonable to expect more at this stage, and I    |
| 43 | am willing to ignore this problem this time. However in the future it will be necessary to see enough       |
| 44 | different sections in each climate period to really understand the rules.                                   |
| 45 | The discussion of how to average the records in order to use the method to its best effect is               |
| 46 | important, but is not very well-explained. I think I got it in the end, and the result is worth discussing, |
| 47 | but I will suggest some better explanation of what was done. I like the thinking in this section though     |
| 48 | - until now it seems to have been assumed that better resolution is always good. Here the authors           |
| 49 | show clearly that better resolution helps with understanding microstructure, but will have to be            |
| 50 | sacrificed to understand large-scale layering.                                                              |
| 51 | Overall, I do see some ways in which the explanations in the paper could be improved. But as a well-        |
| 52 | illustrated proof of concept this is an excellent paper and should be published.                            |
| 53 |                                                                                                             |
| 54 | We thank the referee for the encouraging comments, which we were able to address fully in our               |
| 55 | revision. We have clarified our approach to assessing the spatial significance of single line profiles by   |
| 56 | spatial averaging, aiming to improve readability and to present it in a clearer way. Details are            |
| 57 | presented below. We fully agree with the referee regarding the need for further data in order to            |
| 58 | better assess the significance of the results. This reasoning is also behind the framing of the title,      |
| 59 | where we refer to the datasets as "snapshots". At the present point we believe it was important to          |
| 60 | demonstrate that images from different climatic periods do show distinct differences, and to discuss        |

- 61 how, on this ground, the interpretation of LA-ICP-MS datasets can be improved.

- 64 Detailed comments:
- 65
- 66 Page 1, line 11 "it is demonstrated how instrumental settings can be adapted specifically fit-for-
- 67 purpose". This doesn't quite make sense, I suggest "it is demonstrated how instrumental settings can
- 68 be adapted to be fit-for-purpose".
- 69 Changed accordingly.
- 70
- 71 Line 41. I suspect this became available after the paper was prepared but the authors may wish to
- 72 reference Ng et al 2021 here as well as Rempel et al.
- 73 Changed accordingly.
- 74
- Line 68-71. Like the other reviewer, I didn't understand how one could reach 294 Hz if the washout
- 76 time is 34 ms. Please explain this further.
- 77 We see this potential misunderstanding. We followed state-of-the-art imaging techniques and used a
- 78 dosage of 10 (10 overlapping laser shots per pixel) to improve image quality but did not resolve
- 79 individual pulses this way. We rephrased this in order to separate clearly the general statement
- 80 about the importance of achieving fast washout (line 31) and the specific statement referring to our
- 81 image acquisition (line 74).
  - 30 employing dedicated ablation cells with fast washout as well as optimizing the lasing and ICP-MS settings have introduced a new state-of-the-art in imaging techniques with LA-ICP-MS (Wang et al., 2013; van Elteren et al., 2019). The term "washout time" refers to the time needed to transfer the ablated sample aerosol plume to the ICP-MS. It is principally determined by the extraction efficiency from the ablation cell and any subsequent dispersion in the transfer line. With washout times in the tens of ms range, the recording of baseline-separated single pulses at high repetition rates becomes possible (Van Malderen et al.,
- 82 35 2015). Recently, this new imaging approach was transferred to ice core analysis with LA-ICP-MS, offering the opportunity

ARIS, a rapid aerosol transfer line, was used, resulting in a washout times of ~34 ms. A repetition rate of 294 Hz and a dosage

- 75 of 10 was used here. In contrast to single pulse analysis, a dosage greater than 1 implies that each pixel is generated by multiple partially overlapping laser shots, which leads to an improved signal-to-noise ratio and better image quality (Šala et al., 2021). The fast washout combined with a high repetition rate allows scanning of the surface at around one millimeter per second, which is roughly 10 times faster than previous studies on ice cores (Della Lunga et al., 2017; Spaulding et al., 2017). As a
- 84 Line 75. I don't think you mean 150 mm square! Maybe 150 um? But anyway please be clear whether
- 85 this means 150 x 150, rather than a size that amounts to an area of 150 um2.
- 86 150 x 150. Changed accordingly.
- 87

83

- 88 Figs 2-4. I really like the elemental maps but am a little less clear what I am seeing in the composites
- 89 in part c. Perhaps it's just the colour scale that is confusing me, because superimposing even the
- 90 lightest colours shown there will certainly not give a white. Should the scales run through to very
- 91 light blue/red/green to more correctly characterise what you did?

- 92 The composite images use a standard way of combining chemical channels. We agree with the
- 93 referee that there are some difficulties with this approach, at least as far as using the visual
- 94 inspection for quantitative co-localization investigations. This is a fundamental issue with this way of
- 95 presenting the data, which would not be remedied by using a different color scale. We have referred
- 96 to this in the text already but, following this comment, have added a statement to make it clearer
- 97 (line 144).

where single channel bright spots stand out abundantly. However, the visual overlay has its limitations regarding analyzing colocalizations. Differences in absolute signal intensity and signal/noise ratio can subdue or mask co-localization. The composite
images in Figures 2, 3 and 4 with each element in a separate color channel are thus considered only as a starting point.

- 98
- 99 Line 140. I don't really understand this discussion which leads to the discussion about the use of NIST
- 100 glass reference standards. I can understand that the instrument can be more sensitive to Na, and
- 101 that Na is at higher concentrations so should give higher counts. But I'm not understanding how the
- 102 standards would affect the background or why this is relevant. Do you mean that there is a
- 103 contamination background because of the standard? But then you're clearly seeing a stronger signal
- 104 response as well as a background response for Na. As you can see I am confused so please explain
- 105 what you are suggesting here.
- 106 Following the comments made by both referees, we realize that there was some unintended
- ambiguity in this statement, which we have now rewritten in order to clarify. We only intended to
- 108 refer to the fact that a relatively higher background for Na was observed before in the study by Della
- 109 Lunga et al. (2017) where the NIST glass standards (which we also used) were suggested to be a
- 110 potential cause. As pointed out correctly by the referee, the main issue is however the sensitivity,
- 111 which is also relatively higher for Na, making a clear signal stand out over background. We have
- rewritten this accordingly to clarify it (line 147).

For further comparison of the degree of co-localization, the matrices of intensity values that underlie the images shown in Figure 2, 3 and 4 were used to make scatter plots for each pair of elements. As becomes evident from Figure 5, the intensities for Mg and Sr are generally similar, while Na intensities can be higher by several orders of magnitude. This difference can be explained by higher Na concentrations paired with a higher (absolute) instrumental sensitivity for the element. The scatter

- 150 plots also indicate the almost absent co-localization in the TD Holocene image, showing signs of mutual exclusions (values
- Line 170 and Table 3. Are the elemental ratios in Table 3 ratios by weight or molar ratios?
- 115 The elemental ratios are given as mass ratios (weight), which we have clarified in the text.
- 116
- 117 Around lines 170 and 230: You seem to suggest maybe the marine material is at the grain boundaries
- and the crustal dust in the grains. While this makes sense the ratio of Na/Mg in the grain boundaries
- is much higher than that of sea salt. Might be worth discussion.
- 120 Thank you for pointing this out, we now refer to this observation in the discussion. Our main point in
- 121 this context is that for Mg, we cannot easily distinguish potential sea-salt and dust-related fractions
- 122 based on co-localization analysis with Na and Sr, respectively. Including additional elements may help

- in the future to develop a more sophisticated distinction between marine and crustal material in the
- 124 LA-ICP-MS images. Following careful consideration of the referee's comments, we have re-organized
- 125 the respective section of the discussion (line 240). We believe this will increase the readability
- 126 significantly.
  - 240 The fact that the enrichment at grain boundaries is generally highest for Na, followed by Mg and Sr, suggests that on the micron-scale, differences in the interaction with the grain boundary network exist among these elements and among ice from different climatic periods. Mg may be related to sea salt as well as terrestrial dust (Legrand and Mayewski, 1997). However, based on the LA-ICP-MS images, Mg does not show a clear preference for neither Na (related mostly to sea-salt) nor Sr (a tentative substitute for terrestrial dust sources more commonly investigated through Ca). The Na/Mg ratio also shows the sig-
  - 245 nificant enrichment in Na at the grain boundaries (Table 3). However, it seems worth noting that in the grain interior is within a range typical for sea salt (e.g. Mouri et al., 1993), warrating further investigation.
    Considering the Na enrichment at the grain boundaries in a simplified view would mean that, with grains growing over time, the comparatively mobile (e.g. soluble Na) species are more easily collected at the grain boundaries as opposed to the less mobile species such as the insoluble particulate fraction. This is simplified because particulate inclusions may also inhibit grain
  - 250 boundary growth (e.g. through "pinning of" or "dragging with" grain boundaries). This process could also result in localization of particulate impurities at boundaries (Faria et al., 2014b; Stoll et al., 2021). It is evident that only limited generalized conclusions can be drawn from the small-sized images. Accordingly, it is not intended here to discuss in detail the different behavior of chemical impurities in relation to their mobility and insoluble fractions.
  - However, in future multi-elemental images such a type of analysis may become possible. Imaging the localization of impurities
    does not require a fully quantitative method for this purpose. As an additional indicator, the ratio of intensities, either between boundaries and interiors, or among two elemental species, can also be investigated without calibration. Since LA-ICP-MS measures the total impurity amount, and thus cannot directly distinguish soluble and insoluble fractions, a broader spectrum of elements could serve to identify impurities associated with a specific aerosol based on their glacio-chemical signature (Oyabu et al., 2020).
  - 260 Until images comprising a larger number of elements become available, introducing image analysis techniques can provide an alternative to overcome such limitations. This approach was explored here to compare intra-grain vs. in-grain signals. It is worth pointing out that this type of analysis using image segmentation was performed as a post-processing step and did not require a separate experimental design. Experiments aimed at comparison of intra-grain vs. in-grain were previously performed with LA-ICP-MS but required the manual tracking of the grain boundaries with the laser scan (Beers et al., 2020; Kerch,
  - 265 2016). It becomes clear that the new LA-ICP-MS imaging technique can offer important insights into the ice stratigraphy on the micron-scale and that special merit comes from introducing techniques of image analysis applied to investigating the chemical images. Future efforts in combining techniques image analysis in an automated way and for even larger images seem highly intriguing in this context (Bohleber et al., 2021).

The LA-ICP-MS chemical imaging may offer special merit to investigate the conditions in very deep ice, in particular regarding
impurity diffusion and post-depositional chemical reactions. The localization of the impurities at grain boundaries and triple junctions is a prerequisite for their potential migration along the ice vein network (Rempel et al., 2001; Ng, 2021). The imaging

127

Page 11. I found it really hard to follow what the correlation matrices in Figs 7-9 are. I think I got it in

- 129 the end but please spell it out. If I have understood correctly you have taken all the parallel vertical
- profiles (ie at 420 um resolution you'd have 10 parallel profiles) and correlated them against all the
- 131 others. This should then lead to a symmetrical pattern where perfect correlations would be white
- across the entire diagram. Please explain it in these kind of simple terms. I think it's harder to grasp
- because you have put the figures as rectangles rather than squares, leading the reader to think they
- 134 might be looking at a map, and also to the plot not looking symmetrical.
- 135 This is correct. However, following this comment we have re-written the respective paragraph to
- 136 clarify it in more simple terms (line 195). We are also now using square plots for the correlation
- 137 coefficient and have also updated the supplementary material.

195 Using a gaussian filter along the scan direction in each line mimics the combined effects of increasing washout time and the moving laser (firing at a fixed repetition rate). This is not needed in the transversal direction since individual lines are essentially independent samples. In order to assess the spatial significance of a single longitudinal line, all lines in the image are correlated against each other. The correlation matrix (using the PCC) between all lines in the image is thus symmetric and should be perfectly white (i.e. equal to unity) in case of identical lines. This ideal case would correspond to perfect spatial significance, because it would be irrelevant at which position the individual line profile is measured. The actual images do not fulfill this ideal case. The relative standard deviation (RSD) of the correlation matrix entries is reported to quantify the degree of inhomogeneity.

Figure 7. Example images illustrating the effect of decreasing the spatial resolution of the original image (a) in 35  $\mu$ m steps in the vertical and horizontal direction (see text). The correlation matrix is calculated from all lines in the sub-sampled images, together with its relative standard deviation (RSD). Shown here are results for the TD Holocene Na image, at steps of 210, 420 and 700  $\mu$ m, in tile (b), (c), (d), respectively.